# Bacterial Augmented Floating Treatment Wetlands for Efficient Treatment of Synthetic Textile Dye Wastewater

**Neeha Nawaz [1], Shafaqat Ali [1,2,\*], Ghulam Shabir [3], Muhammad Rizwan [1], Muhammad Bilal Shakoor [1], Munazzam Jawad Shahid [1], Muhammad Afzal [3,\*], Muhammad Arslan [4], Abeer Hashem [5,6], Elsayed Fathi Abd_Allah [7], Mohammed Nasser Alyemeni [5] and Parvaiz Ahmad [5,8]**

[1] Department of Environmental Sciences and Engineering, Government College University, Faisalabad 38000, Pakistan; neehanawaz66@gmail.com (N.N.); mrazi1532@yahoo.com (M.R.); bilalshakoor88@gmail.com (M.B.S.); munazzam01@gmail.com (M.J.S.)

[2] Department of Biological Sciences and Technology, China Medical University, Taichung 40402, Taiwan

[3] National Institute of Biotechnology and Genetic Engineering, Soil and Environmental Biotechnology Division, Faisalabad 38000, Pakistan; gshabirnlbge@yahoo.com

[4] Department of Civil and Environmental Engineering, University of Alberta, Edmonton, AB T6G 2R3, Canada; marslan@ualberta.com

[5] Botany and Microbiology Department, College of Science, King Saud University, P.O. Box. 2460, Riyadh 11451, Saudi Arabia; habeer@ksu.edu.sa (A.H.); mnalyemeni@gmail.com (M.N.A.); parvaizbot@yahoo.com (P.A.)

[6] Mycology and Plant Disease Survey Department, Plant Pathology Research Institute, ARC, Giza 12511, Egypt

[7] Plant Production Department, College of Food and Agricultural Sciences, King Saud University, P.O. Box. 2460, Riyadh 11451, Saudi Arabia; eabdallah@ksu.edu.sa

[8] Department of Botany, S.P. College, Srinagar, Jammu and Kashmir 180001, India

\* Correspondence: shafaqataligill@yahoo.com or shafaqataligill@gcuf.edu.pk (S.A.); manibge@yahoo.com or afzal@nibge.org (M.A.)

**Abstract:** Floating treatment wetland (FTW) is an innovative, cost effective and environmentally friendly option for wastewater treatment. The dyes in textile wastewater degrade water quality and pose harmful effects to living organisms. In this study, FTWs, vegetated with *Phragmites australis* and augmented with specific bacteria, were used to treat dye-enriched synthetic effluent. Three different types of textile wastewater were synthesized by adding three different dyes in tap water separately. The FTWs were augmented with three pollutants degrading and plant growth promoting bacterial strains (i.e., *Acinetobacter junii* strain NT-15, *Rhodococcus sp.* strain NT-39, and *Pseudomonas indoloxydans* strain NT-38). The water samples were analyzed for pH, electrical conductivity (EC), total dissolved solid (TDS), total suspended solids (TSS), chemical oxygen demand (COD), biological oxygen demand (BOD), color, bacterial survival and heavy metals (Cr, Ni, Mn, Zn, Pb and Fe). The results indicated that the FTWs removed pollutants and color from the treated water; however, the inoculated bacteria in combination with plants further enhanced the remediation potential of floating wetlands. In FTWs with *P. australis* and augmented with bacterial inoculum, pH, EC, TDS, TSS, COD, BOD and color of dyes were significantly reduced as compared to only vegetated and non-vegetated floating treatment wetlands without bacterial inoculation. Similarly, the FTWs application successfully removed the heavy metal from the treated dye-enriched wastewater, predominately by FTWs inoculated with bacterial strains. The bacterial augmented vegetated FTWs, in the case of dye 1, reduced the concentration of Cu, Ni, Zn, Fe, Mn and Pb by 75%, 73.3%, 86.9%, 75%, 70% and 76.7%, respectively. Similarly, the bacterial inoculation to plants in the case of dye 2 achieved 77.5% (Cu), 73.3% (Ni), 83.3% (Zn), 77.5% (Fe), 66.7% (Mn) and 73.3% (Pb) removal rates. Likewise in the case of dye 3, which was treated with plants and inoculated bacteria, the metals removal rates were 77.5%, 73.3%, 89.7%,

81.0%, 70% and 65.5% for Cu, Ni, Zn, Fe, Mn and Pb, respectively. The inoculated bacteria showed persistence in water, in roots and in shoots of the inoculated plants. The bacteria also reduced the dye-induced toxicity and promoted plant growth for all three dyes. The overall results suggested that FTW could be a promising technology for the treatment of dye-enriched textile effluent. Further research is needed in this regard before making it commercially applicable.

**Keywords:** floating treatment wetlands; bio-augmentation; dye degradation; bacteria; *Phragmites australis*

## 1. Introduction

Industrialization is a main source of water pollution. The negative impact of polluted water is more severe in developing countries as compare to developed nations [1]. Textile wastewaters contain dyes, and these dyes are one of the worst polluters of our environment [2]. Almost 17% to 20% of industrial water pollution is due to textile dyeing and finishing treatments given to fabrics [3]. Many dyes are derived from heavy metals such as copper (Cu), lead (Pb) and cadmium (Cd). The uses of these metal-complex dyes is a source of heavy metals contamination in water bodies [4]. The release of textile wastewater into open waters causes oxygen level depletion. Dyes block the sunlight in water bodies, thus stopping photosynthesis [5]. These textile contaminants are also carcinogenic and mutagenic for all life forms [3].

Some plants have the capacity to take up pollutants from the environment into themselves [6]. In the past, many plant species have proved to remove or degrade dyes, such as *Sesuvium portulacastrum* that removed Green HE4B, *Portulaca grandiflora* that removed Navy blue HD2R, *Brassica juncea* that removed methyl orange and *Glandularia pulchella* that removed green HE4B [7–9]. Bacteria has the potential to remove dyes from wastewater [10]. Bacteria can also degrade synthetic dyes and use them as a sole source of carbon and energy [11]. There are many examples like degradation of crystal violet by *Enterobacter sp. CV-S1* [12].

Wetland technology has emerged as a sustainable approach for wastewater treatment as compared to conventional treatment processes [13–15]. Floating treatment wetland (FTW) is a variant of pond and wetland land technology (Figure 1), that has been proven as an innovative tool for wastewater treatment [16]. In FTWs, plants are vegetated on an artificial floating mat, such that their roots are submerged in the contaminated water and the aerial parts of the plants remain above the water [13]. The mat can be made of PVC pipes, polyethylene or any other suitable material that can support plants on a water surface [13,17]. Roots play an integral role in and provide space for biofilm formation [16]. Organic matter and other pollutants like heavy metals are taken up by the plants' roots and eventually degraded by bacteria inside the plants and on the roots' surface [11,18]. The roots of plants also act as biological filters as they help in filtration, sedimentation and adsorption of organic matter and suspended particles, as well as other pollutants [19]. In contrast to conventional wetlands, floating wetlands can be installed on any aquatic pond without digging, earth moving and additional land acquisition [13].

The application of specific microorganisms in combination with macrophytes in FTW systems is a recent approach to enhance the pollutant removal efficiency of the system [20,21]. Naturally occurring bacteria and fungi reside inside and outside the plant roots and water, and contribute to pollutants removal process [22]. However, these microorganisms may have limited potential to degrade and remove toxic pollutants [23]. To overcome this concern, FTWs can be restorative by appropriate plant–microbe partnerships [24,25]. This plant–bacteria association may be plant–rhizospheric and or plant–endophytic, depending upon the nature of the bacteria and macrophytes [26,27].

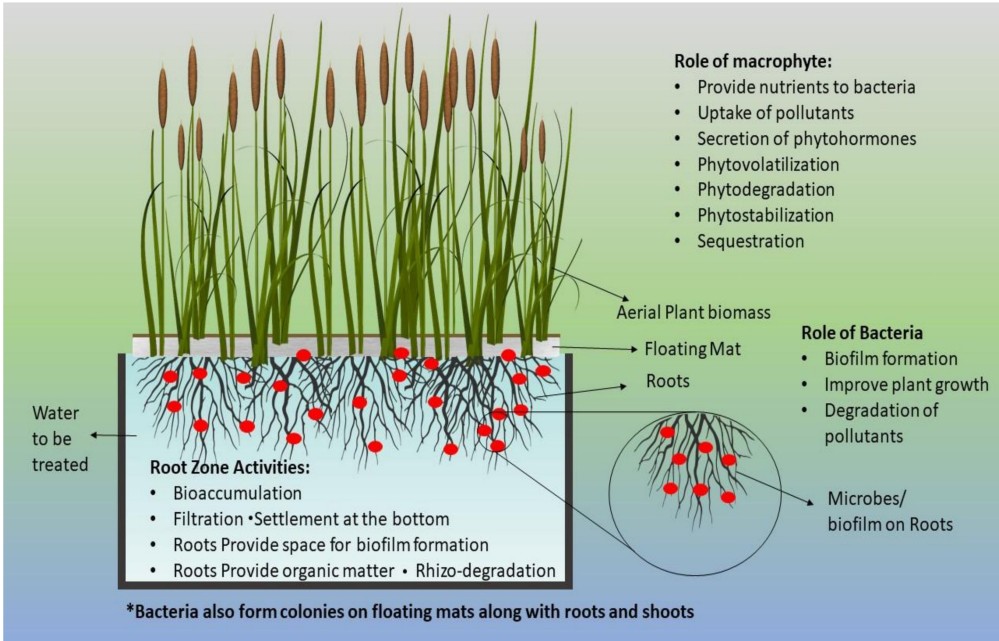

**Figure 1.** Schematic representation of floating treatment wetland and associated pollutant removal process.

Floating wetlands have been widely used for the treatment of wastewater from different sources [28,29]. However, the potential of FTWs composed of *Phragmites australis* in combination with inoculated bacteria has not been fully explored for the treatment of dye-enriched textile effluent. This study was carried out to analyze the potential of *P. australis* and selected bacteria in the degradation of dyes, pollutants reduction and the ultimate alleviation of toxicity of dye enriched water. Further, the focus of this study was on the persistence and survival of inoculated bacteria within the floating wetland system.

## 2. Materials and Methods

### 2.1. Synthesis of Textile Effluent

Three different types of textile effluent were synthesized in the laboratory by mixing three different dyes (500 g) in tap water separately. The first type of effluent contained Bemaplex Navy Blue DRD (D1), the second type of effluent contained Bemaplex Rubine DB (D2) and third one contained Bemaplex Black DRKP Bezma (D3). The concentration of these dyes was 500 mg $L^{-1}$ in each type of synthetic textile effluent. These dyes were selected because of their common use in the textile industry and the high concentration of these toxic dyes and associated degraded products in textile effluent [30]. The experiments were performed individually on each type of effluent.

### 2.2. Macrophytes

The *Phragmites australis* commonly known as common reed was used to carry out this research. It was selected because it has previously proven its effectiveness in reducing the toxicity of polluted wastewater in different studies [11,25,31]. The *P. australis* has an extensive root and shoot system that helps in better oxygen supply to the root zone, thereby enhancing the bacterial propagation and increased pollutants degradation [32].

### 2.3. Endophytic Bacterial Strains

In this study a consortium of three bacterial strains was applied, namely *Rhodococcus sp.* (NCBI Accession: MF326802), *Pseudomonas indoloxydans* (NCBI Accession: MF478985) and *Acinetobacter junii*

(NCBI Accession: MF478980) [25,30]. The strain *P. indoloxydans* was endophyte because it was isolated from the root interior of *Polygonum aviculare*. The strain *Rhodococcus sp.* was rhizospheric as it was isolated from the rhizosphere of *Poa labillardierei*, and the strain *A. junii* was isolated from activated sludge [28]. These specific bacterial strains were chosen due to their potential to reduce textile dyes and assist the macrophytes to alleviate pollutant-induced toxicity without compromising plant growth and development.

The bacterial strains were cultured as separate cultures at 30 °C for 24 h in Luria–Bertani (LB) broth. The bacterial cell pellets were isolated by centrifugation at 4 °C, followed by resuspension in 0.9% NaCl solution [25]. The optical density of each bacterial inoculum was adjusted to 0.9 at 600 nm according to the guidelines of the turbid metric method [33]. The bacterial consortium ($10^8$ colony forming units (CFU) mL$^{-1}$) was prepared by mixing all bacterial inoculum together in equal proportion. This bacterial consortium was used as an inoculum to inoculate the floating treatment wetlands.

## 2.4. Fabrication of FTWs and Experimental Setup

The macrocosms experimental setup was comprised of nine tanks with 1000 liter capacity each, and the dimensions were 1.2 m (L × W × H). The tanks were painted black form all sides to minimize the algal growth. The floating mats were fabricated from expanded polystyrene (EPS)-based sheets manufactured by Diamond® Foam Private Ltd., Pakistan [11,34–36]. EPS sheets are rigid, have low thermal conductivity, are moisture resistant and consist of non-porous closed cell foam [37]. The size of the floating mats was adjusted so that they could float in each tank with >95% coverage on the water surface. All four sides of the floating mats were wrapped with aluminum foil to protect the sheets from sun and water damage. In each floating sheet eight equidistant holes, equal in diameter, were made for the plantation of macrophytes on the floating mats. Each hole was planted with three healthy seedlings of *P. australis*, thus having 24 seedlings in each mat. Each seedling weighed 45 to 65 g and their length was 55–65 cm. The seedlings were supported by coconut shavings and soil in the floating mat. The seedlings were allowed to grow in fresh water for one month to gain optimum growth of roots and shoots. After one month, the average height of the plants was about 145 cm, and the fresh water in tanks was replaced with the synthetic textile effluent enriched with dyes. The experiment was run in triplicate with the subsequent treatment design:

T1D1, T1D2, T1D3: Only dye;

T2D1, T2D2, T2D3: Containing dye and plants;

T3D1, T3D2, T3D3: Dye, plants and bacterial consortium;

T4: Fresh water and plants.

(D1: Bemaplex Navy Blue DRD, D2: Bemaplex Rubine DB, D3: Bemaplex Black DRKP Bezma).

The treatments T3D1, T3D2, T3D3 were inoculated by pouring one liter of inoculum into each tank. The experiment lasted for 20 days until a maximum of dye and pollutants were removed from treated water. One liter of sample was collected from each tank every 5 days starting from day 0 using a sequencing fill-and-draw batch mode method (for convenience, data of only the 0, 10th and 20th day are presented in results). The samples were stored in a cool and dry place for further analysis [38]. The collected water samples were analyzed for pH, electrical conductivity (EC), total dissolved solids (TDS), total suspended solids (TSS), dye concentration, chemical oxygen demand (COD), biological oxygen demand (BOD), colony forming unit (CFU) and metal concentration (Cu, Fe, Mn, Ni, Zn and Pb) according to standard methods [38]. The evapo-transpiration losses were recovered by pouring fresh water in treatment tanks up to the level of 1000 L in each tank [34]. In case of rain, the tanks were covered with plastic sheets.

## 2.5. Persistence of Inoculated Bacteria in Treated Water and Plants

The persistence of bacteria in water, root and shoot samples were periodically analyzed during the experiment using the cultivation-dependent plate count method [24,25]. The collected roots and shoots samples were surface sterilized by 70% ethanol and 2% sodium hypochlorite solution. Then these roots

and shoots were homogenized in a 0.9% NaCl solution and serial dilution of these suspensions was spread on LB agar plates. Similarly, the collected water samples from all treatments were spread on LB agar plates and these plates were incubated at 37 °C for 48 h for CFU analysis [35,39].

*2.6. Plant Biomass*

In order to determine the effect of bacterial inoculation and dye-induced toxicity on plants growth and development, the data about plants agronomic parameters (root and shoot length and dry biomass) were noted at the end of the experiment. The root and shoot length was measured manually by a measuring scale. The root and shoots were harvested near the surface of the floating mat and oven dried at the 80 °C for 72 h until a constant weight was achieved [11,34].

*2.7. Statistical Analysis*

The results of physicochemical parameters (pH, EC, TDS, TSS, BOD, COD, color and heavy metals), bacterial persistence and plant biomass were evaluated by the SPSS software package. The comparison between treatments was executed by analysis of variance (ANOVA) followed by a Post-Hoc Tukey test ($p \leq 0.05$) [40]. The alphabet labels over the values show the significant/non-significant differences among treatments.

## 3. Results

*3.1. Changes in Physicochemical Parameters of Treated Textile Effluent*

The graphs in Figures 2–4 represent the changes in the physicochemical parameters of the dye-enriched tap water treated by floating treatment wetlands. The floating wetlands had a positive impact and predominately reduced the pH, EC, TSS, TDS, COD, BOD and color within the retention period of 20 days. All of the above-mentioned pollutants were reduced sharply in vegetated treatments (T2D1, T2D2, T2D3 and T3D1, T3D2, T3D3) as compared to non-vegetated treatments. However, the vegetated treatments inoculated with bacterial consortium (T3D1, T3D2, T3D3) achieved highest pollutants removal rate, outperforming all other treatments in all three types of dyes.

In the treatment containing dye 1, *P. australis* and bacterial consortium (T3D1), maximum pollutants removal efficiency was achieved. In this treatment, pH was reduced to 6.7 from 8.5, EC was reduced from 6.13 to 1.00 mS cm$^{-1}$, TDS was reduced from 400 to 60 mg L$^{-1}$, TSS was reduced from 92 to 19 mg L$^{-1}$, COD was reduced from 310 to 30 mg L$^{-1}$, BOD was reduced from 121 to 20 mg L$^{-1}$ and color was reduced from 40.0 to 6.0 m$^{-1}$.

Similarly, in the case of dye 2, maximum pollutant removal efficiency was obtained from T3D2, in which pH was reduced to 6.8 from 8.5, EC was reduced from 6.13 to 1.02 mS cm$^{-1}$, TDS was reduced from 400 to 63 mg L$^{-1}$, TSS was reduced from 92 to 21 mg L$^{-1}$, COD was reduced from 308 to 33 mg L$^{-1}$, BOD was reduced from 121 to 18 mg L$^{-1}$ and color was reduced from 40.0 to 6.7 m$^{-1}$.

As in the case of dye 1 and dye 2, the maximum pollutant removal rate was achieved by T3D3 containing dye 3, *P. australis* and bacterial consortium. In this treatment, pH was reduced to 6.7 from 8.5, EC was reduced from 6.15 to 1.05 mS cm$^{-1}$, TDS was reduced from 401 to 62 mg L$^{-1}$, TSS was reduced from 91 to 24 mg L$^{-1}$, COD was reduced from 309 to 31 mg L$^{-1}$, BOD was reduced from 120 to 19 mg L$^{-1}$ and color was reduced from 40.0 to 6.4 m$^{-1}$.

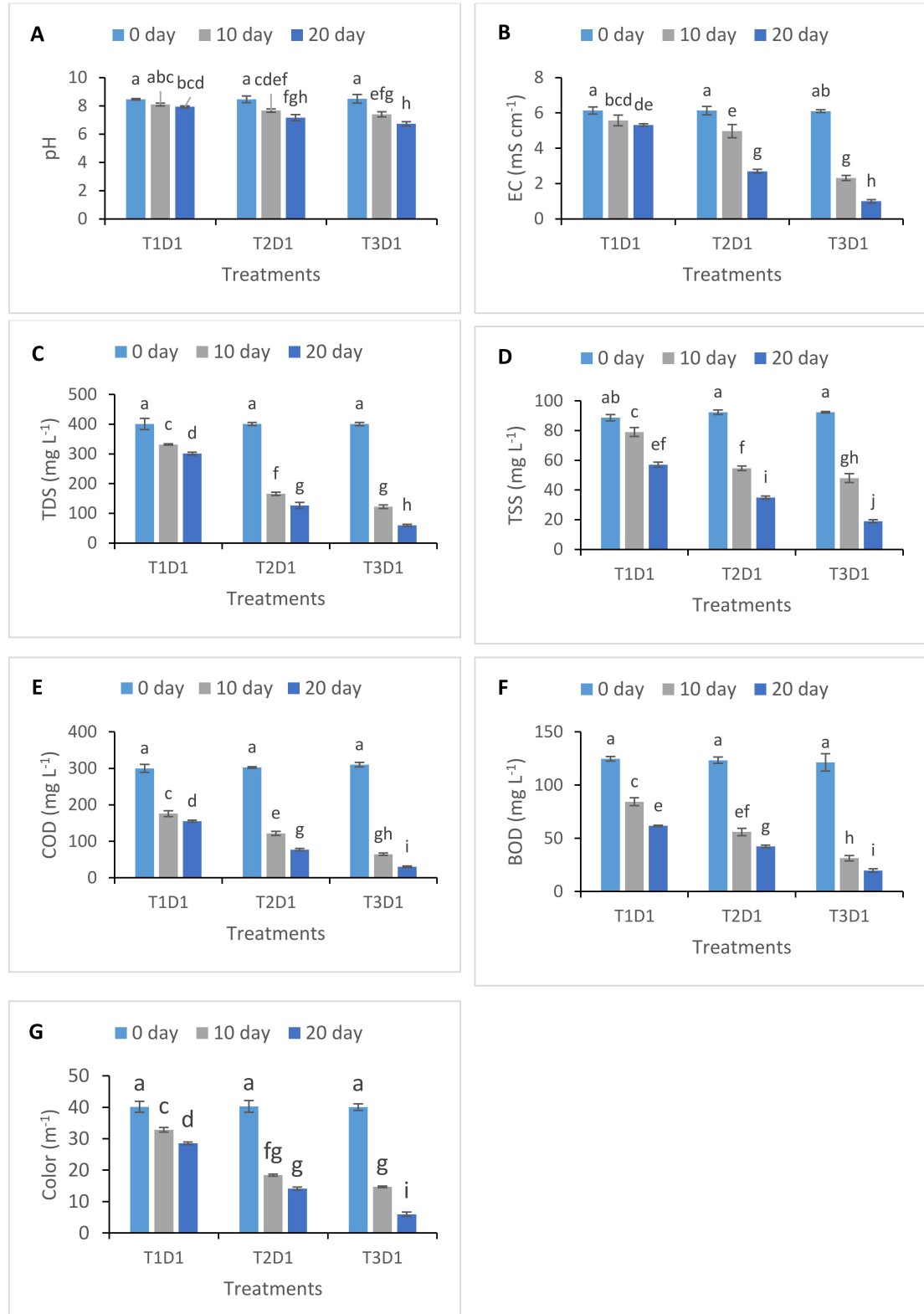

**Figure 2.** Effect of floating treatment wetlands on pH (**A**), EC (**B**), TDS (**C**), TSS (**D**), COD (**E**), BOD (**F**) and color (**G**) after 20 days of retention time. D1: Bemaplex Navy Blue DRD. Each value is a mean of three replicates and error bars represent the standard deviation. Lettering shows that various treatments are significantly different at $p \leq 0.05$.

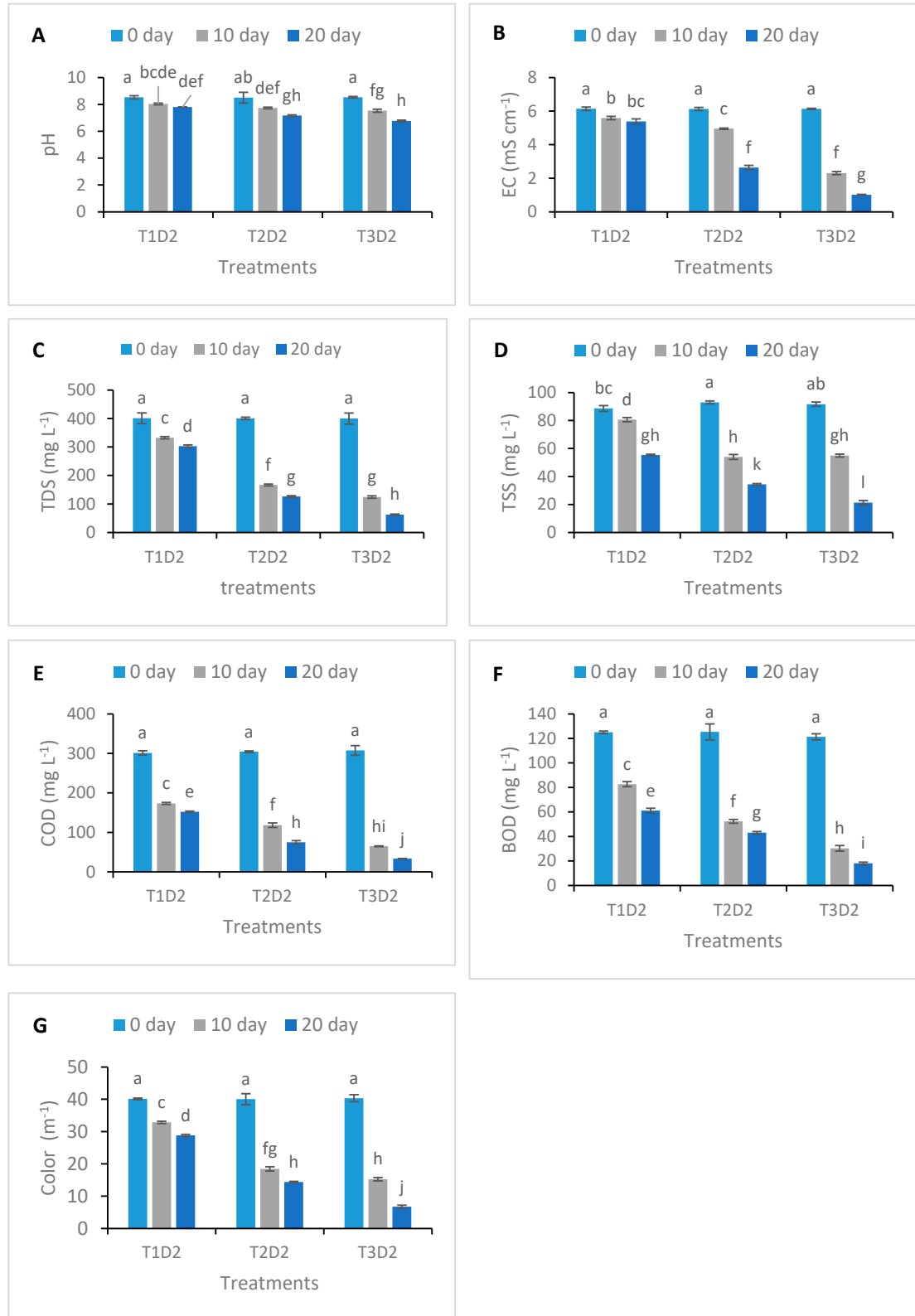

**Figure 3.** Effect of floating treatment wetlands on pH (**A**), EC (**B**), TDS (**C**), TSS (**D**), COD (**E**), BOD (**F**) and color (**G**) after 20 days of retention time. D2: Bemaplex Rubine DB. Each value is a mean of three replicates and error bars represent the standard deviation. Lettering shows that various treatments are significantly different at $p \leq 0.05$.

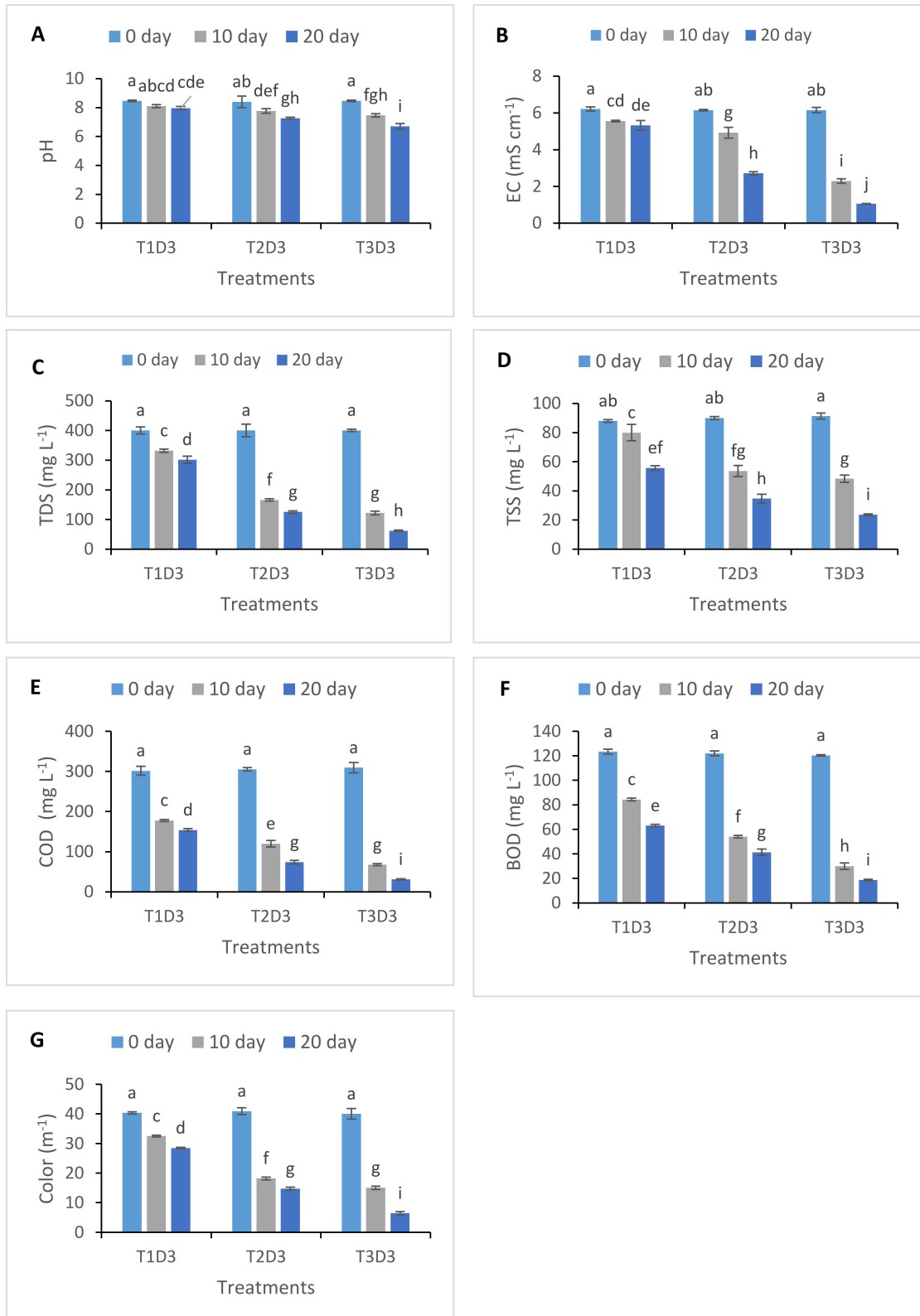

**Figure 4.** Effect of floating treatment wetlands on pH (**A**), EC (**B**), TDS (**C**), TSS (**D**), COD (**E**), BOD (**F**) and color (**G**) after 20 days of retention time. D3: Bemaplex Black DRKP Bezma. Each value is a mean of three replicates and error bars represent the standard deviation. Lettering shows that various treatments are significantly different at $p \leq 0.05$.

## 3.2. Removal of Heavy Metals from Water

The concentration of six heavy metals (Cu, Fe, Mn, Ni, Zn and Pb) considerably reduced in the FTWs-treated water samples (Table 1). All vegetated treatments (T2 and T3) showed significantly better removal of trace metals from the dye-polluted water (D1, D2 and D3) as compared to the non-vegetated treatments (T1D1, T1D2 and T1D3). Next, the efficiency of bacterial augmented treatments (T3D1, T3D2 and T3D3) was significantly better than non-inoculated vegetated treatments (T2D1, T2D2 and T2D3). In treatment T3D1, the metal concentrations for Cu, Ni, Zn, Fe, Mn and Pb were reduced by up to 75%, 73.3%, 86.9%, 75%, 70% and 76.7%, respectively, in the 20 days retention time. Similar results were achieved for dye 2 and dye 3 in the case of treatment T3, in which bacterial inoculation efficiently removed the metals from dye water as compared to non-inoculated vegetated treatments (T2) and un-vegetated non-inoculated treatments (T1).

**Table 1.** Percentage (%) reduction in concentration of metals with time by floating treatment wetlands.

| Treatment | | T1 Only Dye | | T2 Dye + Plant | | T3 Dye + Plant + Bacteria | |
|---|---|---|---|---|---|---|---|
| Dye | Metals | 10 Days | 20 Days | 10 Days | 20 Days | 10 Days | 20 Days |
| D1 | Cu | 20.0 [b,c] (0.0) | 30.0 [c,d] (0.0) | 58.5 [e] (0.0) | 65.9 [e,f,g] (0.0) | 67.5 [e,f,g] (0.0) | 75.0 [g] (0.0) |
| | Ni | 19.4 [b,c] (0.0) | 32.3 [d,e] (0.0) | 40.0 [e,f] (0.0) | 60.0 [g,h] (0.0) | 60.0 [g,h] (0.0) | 73.3 [h] (0.0) |
| | Zn | 8.8 [c] (0.0) | 21.1 [d] (0.0) | 60.0 [e,f] (0.0) | 66.7 [f] (0.0) | 75.4 [g] (0.0) | 86.9 [h] (0.0) |
| | Fe | 7.5 [b,c] (0.0) | 12.5 [c] (0.0) | 48.8 [e] (0.0) | 65.9 [g] (0.0) | 62.5 [f,g] (0.0) | 75.0 [h] (0.0) |
| | Mn | 13.3 [b,c] (0.0) | 23.3 [c,d] (0.0) | 34.5 [d,e] (0.0) | 48.3 [e,f] (0.0) | 56.7 [f,g] (0.0) | 70.0 [h] (0.0) |
| | Pb | 20.0 [b,c] (0.0) | 26.7 [c,d] (0.0) | 40.0 [d,e,f] (0.0) | 60.0 [g] (0.0) | 46.7 [f] (0.0) | 76.7 [h] (0.0) |
| D2 | Cu | 20.0 [c] (0.0) | 27.5 [c] (0.0) | 55.0 [d,e,f] (0.0) | 65.0 [f,g,h] (0.0) | 70.0 [g,h,i] (0.0) | 77.5 [i] (0.0) |
| | Ni | 16.7 [b,c] (0.0) | 30.0 [c] (0.0) | 46.7 [d] (0.0) | 60.0 [d,e,f] (0.0) | 60.0 [d,e,f] (0.0) | 73.3 [f] (0.0) |
| | Zn | 13.3 [b,c] (0.0) | 33.3 [c] (0.0) | 56.9 [d,e] (0.0) | 65.5 [e,f] (0.0) | 75.0 [f,g] (0.0) | 83.3 [g] (0.0) |
| | Fe | 5.1 [a,b] (0.0) | 12.8 [b,c] (0.0) | 48.8 [c,d] (0.0) | 65.9 [e,f] (0.0) | 62.5 [e] (0.0) | 77.5 [f] (0.0) |
| | Mn | 22.6 [b] (0.0) | 25.8 [b,c] (0.0) | 40.0 [c,d,e] (0.0) | 50.0 [e,f,g] (0.0) | 60.0 [f,g] (0.0) | 66.7 [g] (0.0) |
| | Pb | 22.6 [b] (0.0) | 29.0 [b,c] (0.0) | 43.3 [c,d,e] (0.0) | 56.7 [e,f,g] (0.0) | 50.0 [e,f] (0.0) | 73.3 [g] (0.0) |
| D3 | Cu | 20.0 [a,b,c] (0.0) | 30.0 [a,b,c,d,e] (0.0) | 55.0 [d,e,f,h] (0.0) | 65.0 [f,g] (0.0) | 67.5 [e,f,g] (0.0) | 77.5 [g] (0.0) |
| | Ni | 20.0 [c,d] (0.0) | 30.0 [d,e] (0.0) | 43.8 [e,f] (0.0) | 59.4 [g,h] (0.0) | 60.0 [g,h,i] (0.0) | 73.3 [i] (0.0) |
| | Zn | 10.5 [c] (0.0) | 24.6 [d] (0.0) | 58.3 [e] (0.0) | 70.0 [f,g,h] (0.0) | 75.9 [g,h] (0.0) | 89.7 [i] (0.0) |
| | Fe | 14.6 [b,c] (0.0) | 19.5 [c] (0.0) | 52.4 [d,e] (0.0) | 69.0 [f,g] (0.0) | 66.7 [e,f] (0.0) | 81.0 [g] (0.0) |
| | Mn | 10.0 [a,b,c] (0.0) | 26.7 [c,d,e] (0.0) | 35.7 [d,e,f] (0.0) | 46.4 [f,g,h] (0.0) | 63.3 [g,h] (0.0) | 70.0 [h] (0.0) |
| | Pb | 23.3 [bc] (0.0) | 30.0 [cd] (0.0) | 43.3 [ef] (0.0) | 56.7 [hi] (0.0) | 44.8 [fg] (0.0) | 65.5 [i] (0.0) |

T symbolizes treatments (T1, T2, T3) and D symbolizes dye (D1: Bemaplex Navy Blue DRD, D2: Bemaplex Rubine DB, D3: Bemaplex Black DRKP Bezma). Values represent the means of three replicates and standard deviations are presented in parenthesis. Lettering shows that various treatments are significantly different at $p \leq 0.05$.

### 3.3. Bacterial Persistence in Roots, Shoots and Water

The presence of a significantly high population of bacteria in water (Table 2), roots and shoots (Table 3) in the bacterial inoculated treatment (T3) as compared to non-inoculated treatments (T1 and T2) confirmed the persistence of inoculated bacteria during the treatment process in inoculated treatments for all three dyes. The bacteria showed the highest population in wastewater compared to roots and shoots. On the other hand, the count of bacteria was found higher in roots than shoots.

**Table 2.** Average concentration of bacteria in water (colony forming unit (CFU) mL$^{-1}$).

| Treatment | Days | Dye 1 | Dye 2 | Dye 3 |
|---|---|---|---|---|
| Only Dye (T1) | 5 | $1.5 \times 10^3$ [a] (0.3) | $1.6 \times 10^3$ [a] (0.2) | $1.5 \times 10^3$ [a] (0.4) |
| | 10 | $1.6 \times 10^3$ [a] (0.4) | $1.6 \times 10^3$ [a] (0.5) | $1.7 \times 10^3$ [a] (0.5) |
| | 15 | $1.9 \times 10^3$ [a] (0.6) | $1.8 \times 10^3$ [a] (0.7) | $1.8 \times 10^3$ [a] (0.6) |
| | 20 | $1.8 \times 10^3$ [a] (0.6) | $1.8 \times 10^3$ [a] (0.8) | $2.0 \times 10^3$ [a] (0.9) |
| Dye + Plant (T2) | 5 | $2.1 \times 10^5$ [b] (1.0) | $2.3 \times 10^5$ [b] (1.0) | $2.2 \times 10^5$ [b] (0.9) |
| | 10 | $2.7 \times 10^5$ [b] (1.1) | $2.9 \times 10^5$ [b] (1.2) | $2.5 \times 10^5$ [b] (1.1) |
| | 15 | $3 \times 10^5$ [b,c] (0.9) | $3.3 \times 10^5$ [b,c] (0.8) | $3.4 \times 10^5$ [b,c] (0.9) |
| | 20 | $3.7 \times 10^5$ [c] (1.1) | $3.5 \times 10^5$ [c] (1.1) | $3.6 \times 10^5$ [c] (1.1) |
| Dye + Plant + Bacteria (T3) | 5 | $9.6 \times 10^8$ [d] (0.5) | $9.8 \times 10^8$ [d] (0.6) | $9.9 \times 10^8$ [d] (0.5) |
| | 10 | $7.1 \times 10^9$ [e] (0.7) | $7.2 \times 10^9$ [e] (0.5) | $7.1 \times 10^9$ [e] (0.4) |
| | 15 | $6.4 \times 10^9$ [f] (0.2) | $6.6 \times 10^9$ [f] (0.6) | $6.6 \times 10^9$ [f] (0.6) |
| | 20 | $5.0 \times 10^8$ [g] (0.6) | $5.1 \times 10^8$ [g] (0.6) | $4.9 \times 10^8$ [g] (0.7) |

Dye 1: Bemaplex Navy Blue DRD, Dye 2: Bemaplex Rubine DB, Dye 3: Bemaplex Black DRKP Bezma. Values represent the means of three replicates and standard deviations are presented in parenthesis. Lettering shows that various treatments are significantly different at $p \leq 0.05$.

### 3.4. Plant Growth in Response to Bacterial Inoculation

It is well established that the presence of toxic pollutants in water inhibits plant growth and ultimately phytoremediation efficiency. The root and shoot length (Table 4) and root and shoot dry mass (Table 5) were noted at the end of the experiment and it was found that the plants grown in dye water inoculated with bacteria (T3) showed more growth as compared to the plants grown only in dye water. The plants grown in only tap water with no dye showed maximum growth out of all treatments. The dye water hindered the growth of plants and root and shoot length were reduced in case of all three dyes. Similarly, the plants grown in dye water inoculated with bacteria gained high shoot and root dry biomass due to good growth as compared to plants grown in dye water without bacterial inoculation. These results showed that despite the toxic effect of dyes, the inoculation of bacteria to dye water predominantly increased the length and dry weight of shoot and root of *P. australis*.

**Table 3.** Average concentration of bacteria in roots and shoots (CFU mL$^{-1}$).

| Root/Shoot | Treatment | Days | Dye 1 | Dye 2 | Dye 3 |
|---|---|---|---|---|---|
| Root | Dye + Plant (T2) | 5 | $2 \times 10^{2 a}$ (0.8) | $2.2 \times 10^{2 a}$ (0.9) | $2.2 \times 10^{2 a}$ (0.8) |
| | | 10 | $3.2 \times 10^{2 b}$ (1.0) | $3.3 \times 10^{2 b}$ (1.0) | $3.3 \times 10^{2 b}$ (1.0) |
| | | 15 | $3.7 \times 10^{2 b,c}$ (0.8) | $3.7 \times 10^{2 b,c}$ (0.8) | $3.7 \times 10^{2 b,c}$ (0.8) |
| | | 20 | $4.1 \times 10^{2 c}$ (0.9) | $4.0 \times 10^{2 c}$ (0.9) | $4.1 \times 10^{2 c}$ (0.9) |
| | Dye + Plant + Bacteria (T3) | 5 | $4 \times 10^{3 d}$ (1.2) | $4.4 \times 10^{3 d}$ (1.3) | $4.3 \times 10^{3 d}$ (0.9) |
| | | 10 | $11.9 \times 10^{3 e}$ (1.1) | $12.0 \times 10^{3 e}$ (1.3) | $11.6 \times 10^{3 e}$ (1.1) |
| | | 15 | $17.2 \times 10^{3 f}$ (1.1) | $17.6 \times 10^{3 f}$ (1.3) | $18.5 \times 10^{3 f}$ (0.9) |
| | | 20 | $22.8 \times 10^{3 g}$ (1.1) | $23.1 \times 10^{3 g}$ (1.1) | $23.4 \times 10^{3 g}$ (1.1) |
| Shoot | Dye + Plant (T2) | 5 | $1.1 \times 10^{2 a}$ (0.2) | $1.2 \times 10^{2 a}$ (0.2) | $1.2 \times 10^{2 a}$ (0.3) |
| | | 10 | $1.2 \times 10^{2 a}$ (0.4) | $1.0 \times 10^{2 a}$ (0.5) | $1.2 \times 10^{2 a}$ (0.3) |
| | | 15 | $1.3 \times 10^{2 a}$ (0.2) | $1.3 \times 10^{2 a}$ (0.2) | $1.2 \times 10^{2 a}$ (0.7) |
| | | 20 | $1.3 \times 10^{2 a}$ (0.2) | $1.2 \times 10^{2 a}$ (0.3) | $1.3 \times 10^{2 a}$ (0.5) |
| | Dye + Plant + Bacteria (T3) | 5 | $1.4 \times 10^{3 b}$ (0.3) | $1.6 \times 10^{3 b}$ (0.2) | $1.5 \times 10^{3 b}$ (0.4) |
| | | 10 | $6.2 \times 10^{3 c}$ (0.7) | $6.0 \times 10^{3 c}$ (0.7) | $6.2 \times 10^{3 c}$ (0.8) |
| | | 15 | $10.5 \times 10^{3 d}$ (0.4) | $10.1 \times 10^{3 d}$ (0.9) | $11.2 \times 10^{3 d}$ (1.3) |
| | | 20 | $14.3 \times 10^{3 e}$ (2.1) | $13.9 \times 10^{3 e}$ (2.3) | $14.0 \times 10^{3 e}$ (2.5) |

Dye 1: Bemaplex Navy Blue DRD, Dye 2: Bemaplex Rubine DB, Dye 3: Bemaplex Black DRKP Bezma. Values represent the means of three replicates and standard deviations are presented in parenthesis. Lettering shows that various treatments are significantly different at $p \leq 0.05$.

**Table 4.** Comparison between shoot lengths and root lengths in different treatments.

| Treatments | Shoot Length (cm) | | | Root Length (cm) | | |
|---|---|---|---|---|---|---|
| | Dye 1 | Dye 2 | Dye 3 | Dye 1 | Dye 2 | Dye 3 |
| Dye + Plants (T2) | 187.7 [d] (24.9) | 197.7 [c,d] (7.8) | 202.7 [b,c,d] (3.8) | 29.7 [c] (0.58) | 30.7 [c] (1.2) | 31.0 [c] (0.0) |
| Dye + Plants+ Bacteria (T3) | 222.0 [a,b,c] (10.8) | 228.0 [a,b] (2.6) | 224.3 [a,b,c] (9.3) | 38.0 [b] (1.0) | 39.0 [b] (1.0) | 38.3 [b] (1.2) |
| Fresh water + Plants (T4) | 233.3 [a] (3.1) | 232.3 [a] (2.5) | 230.0 [a,b] (2.6) | 43.0 [a] (1.0) | 44.3 [a] (0.58) | 43.7 [a] (2.1) |

Dye 1: Bemaplex Navy Blue DRD, Dye 2: Bemaplex Rubine DB, Dye 3: Bemaplex Black DRKP Bezma. Values represent the means of three replicates and standard deviations are presented in parenthesis. Lettering shows that various treatments are significantly different at $p \leq 0.05$.

**Table 5.** Shoot and root dry weight of the plants.

| Treatments | Shoot Dry Weight (g) | | | Root Dry Weight (g) | | |
|---|---|---|---|---|---|---|
| | Dye 1 | Dye 2 | Dye 3 | Dye 1 | Dye 2 | Dye 3 |
| Dye + Plants (T2) | 492.0 [c] (65.3) | 517.7 [b,c] (20.3) | 533.0 [a,b,c] (9.6) | 62.3 [c] (1.2) | 64.3 [c] (3.2) | 64.7 [c] (0.6) |
| Dye + Plants+ Bacteria (T3) | 572.7 [a,b] (29.3) | 591.3 [a,b] (7.1) | 581.3 [a,b] (21.4) | 79.3 [a,b,c] (2.1) | 81.0 [b,c] (1.7) | 80.3 [a,b,c] (2.1) |
| Fresh water + Plants (T4) | 605.0 [a] (8.2) | 609.0 [a] (9.5) | 603.7 [a] (8.3) | 89.3 [a,b] (3.1) | 93.0 [a] (1.7) | 90.0 [a,b] (1.7) |

Dye 1: Bemaplex Navy Blue DRD, Dye 2: Bemaplex Rubine DB, Dye 3: Bemaplex Black DRKP Bezma. Values represent the means of three replicates and standard deviations are presented in parenthesis. Lettering shows that various treatments are significantly different at $p \leq 0.05$.

## 4. Discussion

In this study, pH, EC, TDS, TSS, COD, BOD and color of the dye-enriched water and heavy metals contents were significantly decreased in the vegetated and vegetated-inoculated floating treatment wetlands. The reductions in pollutants load in treated dye-contaminated water emphasize the prominent role of vegetation and bacteria in floating wetlands.

The pH might be decreased due to the release of organic acids by the roots of the plants as reported in earlier studies [31,41]. The decrease in EC might be associated with the uptake of nutrients by plants and the biological and physicochemical binding of pollutants to roots and soil particles [13,36]. The pH and EC reduction was highest in treatment vegetated with plants and augmented with bacteria. This suggests the key role of plants and bacteria in pH and EC reduction through the release of organic acids and the uptake of nutrients by plants and bacteria [25,34,35]. The TDS and TSS loads were reduced due to the combination of physical and biological processes supported by floating wetlands [41]. The suspended particles in the water are trapped in the biofilm of the roots of macrophytes, and there they either precipitate at the bottom or adsorb on biofilm where they might be degraded [42]. Physical entrapment in roots, sorption and settlement at the bottom might contribute to the removal of TDS and TSS from treated water [16,19,43]. Further, the roots of plants act as physical filters and provide appropriate organic matter that acts as a bio-sorbent and contributed to the removal of particulate matter [11,21].

Roots allow microbial communities to assimilate carbon compounds and reduce BOD and COD [44]. In this study, the high removal of BOD in wetland systems might be attributed to the deposition and filtration of organic compounds that can be settled. The speedy and high removal rate in bacterial augmented FTWs could be attributed to the biofilm on roots, which contributes to the removal of organic matter by decomposing it into simple nutrients, thus aiding in the direct uptake by the plant [20,45]. Uptake by plants' roots is an important method of nutrient removal [42]. The nutrients in the wastewater might be taken up by the roots of the plants. There, they can either accumulate in the plant biomass or be degraded by endophytic bacteria present inside the plants [25,46]. The similar findings have been reported by earlier studies, where plants and bacterial combinations enhanced the removal of organic pollutants from highly polluted wastewater [31,47,48].

Color was also removed to a great extent in this study by the vegetated treatment and the vegetated-inoculated treatment. It has been well reported that COD, a measure of oxidizable contaminants, has a positive correlation with color in textile wastewater [11]. Correspondingly, in this study color was reduced with the reduction in COD. However, the rate of decolorization was high in vegetated-inoculated floating wetlands. This could be associated with the combined action of plant and bacteria in the degradation of dyes and removal of color [11]. This emphasized the key role of bacteria in the decolorization of dye from textile effluent. The previous studies also showed that many bacteria are helpful in the removal of dyes, and that bacteria have the ability to degrade dyes by aerobic as well as anaerobic mechanisms [11,49].

In this study, the concentration of six heavy metals (Cr, Ni, Zn, Fe, Pb and Mn) was decreased significantly in the treated dye-containing wastewater. The unique potential of *P. australis* to remove heavy metals has been reported by many researchers [25,34]. In the previous studies, *P. australis* showed similar pattern of removal of heavy metals from industrial effluent [11,28,36]. These previous studies also demonstrated that the heavy metals from wastewater were taken up by the *P. australis* in its roots and shoots [41,50,51]. The maximum concentrations of heavy metals were found in the roots of the plant, meaning that the root has most potential to uptake heavy metals [50].

In the case of inoculation of *P. australis* with bacteria, the heavy metal removal capacity was further enhanced. The improved performance of bacterially augmented FTWs emphasized the role of bacteria in the removal of heavy metals from polluted water. The inoculated bacteria reduced the metals load in polluted water by their bioaccumulation potential [31]. These bacteria might contribute to reducing metal-induced toxicity and increase the bioavailability and metals uptake of plants [27]. It is well reported that in FTWs the inoculated bacteria may boost the metals removal process by entrapment of metals in root biofilms, sorbing of metallic ions on the bacterial cell wall and oxidation of metal ions [52,53]. Further, the plaque formation by the combined action of plant and bacteria on plant roots may increase the Fe, Mn, Cu and Zn binding in roots biofilms [13,54]. This emphasizes that *P.australis* and inoculated bacterial combined role, which contributes to metals removal from treated dye-contaminated wastewater. The significantly substantial removal of metals from bacterial inoculated treatments relative to non-inoculated vegetated treatments could be attributed to a high population of bacteria in the inoculated treatment.

The inoculated bacteria showed persistence in polluted water being treated by floating treatment wetlands. The periodic analysis of water from all three treatments showed the high population of bacteria in inoculated treatments as compared to the non-inoculated treatments. The higher population of bacteria in the water of inoculated treatments confirmed that inoculated bacteria showed persistence and were responsible for dye removal and pollutant removal. This could be due to the fact that the inoculated bacteria successfully made mutualistic relationships with plants, which supported the survival of inoculated bacterial [55]. This finding is consistent with previous studies in which inoculated bacteria improved the pollutant removal process [24,34]. The survival of inoculated bacteria depends upon the nutrient supply, pH, temperature and the interaction with the host [56,57]. In this study, the bacterial population in the roots and shoots of inoculated plants were found to be higher as compared to non-inoculated vegetated treatment. This could be due to the preferential survival of bacteria in roots and shoots of *P. australis* in inoculated treatments, as reported in previous studies [28,58]. Further, these bacteria were initially isolated from the roots and shoots of the plants; hence these bacteria possibly have an adaptive mechanism to survive and grow in these parts of the plant in this hostile environment [18,27]. In order to make FTWs a potential wastewater treatment method, periodic inoculation of bacteria should be performed in order to overcome the problem of decreasing bacteria with time in inoculated water [57,59].

Toxic pollutants in the environment inhibit plant growth [27]. Dyes containing toxic chemicals and potentially toxic heavy metals also inhibit plant growth [28]. In this study, the *P. australis*, synergistic with bacteria, achieved high root and shoot growth as compared to plants without inoculation. The control tank having only water and plants with no added dye showed maximum growth of roots and shots of plants due to the absence of any toxic pollutant. The bacteria present in the system can promote plant growth by decreasing biotic and abiotic stress [60]. Bacteria also positively affect plant growth by releasing phyto-hormones and by the solubilisation of essential nutrients [61]. Pollutant-degrading rhizospheric and endophytic bacteria have been proven as effective to enhance plant growth development and phytoremediation efficacy [53]. Similar results have been reported by previous studies where inoculated bacteria promoted plant growth by alleviating pollutant-induced toxicity and improved plant nutrition, health and disease resistance [34,62].

## 5. Conclusions

The present study evaluated the potential of *P. australis* in FTWs along with three inoculated bacterial strains to remove dye as well as organic and inorganic pollutants from dye-enriched water. The results clearly indicated that *P. australis* along with inoculated strains have a great potential to remove different types of dyes and pollutants, including potentially toxic metals, from textile effluent. The floating wetlands are capable of efficiently decreasing the levels of pH, EC, TDS, TSS, BOD, COD, color and toxic metals from dye-polluted wastewater. The high rate of pollutants removal by vegetated-inoculated FTWs validates the potential role of bacteria in FTWs. The bacteria showed high persistence in water as well as in the roots and shoots of the inoculated plants. It suggests that bacteria have the ability to make a mutualistic relationship with *P. australis* in FTWs system to collectively remove pollutants form the water body. These plant growth-promoting rhizospheric and endophytic bacteria also increased the plants' ability to tolerate pollutant-induced toxicity and alleviate the toxicity of textile effluent. We conclude that the FTWs can be a promising technology to treat textile effluent and can be a propitious substitute for conventional wastewater technology for the treatment of textile effluent. The pollutant removal efficiency of already existing water retention ponds can be enhanced by installing floating wetland systems. However, there is a need for conducting meticulous research about the careful and objective-based selection of plants and bacteria, which can further enhance the efficiency of the FTWs system.

**Author Contributions:** Conceptualization, N.N., S.A., G.S., M.A. (Muhammad Afzal), M.A. (Muhammad Arslan) and M.N.A.; Data curation, N.N. and G.S.; Formal analysis, N.N., M.R. and M.J.S.; Funding acquisition, A.H. and P.A.; Investigation, S.A., M.A. (Muhammad Arslan) and P.A.; Methodology, N.N., G.S., M.B.S., M.J.S., M.A. (Muhammad Afzal) and A.H.; Project administration, S.A., M.R. and M.A. (Muhammad Afzal); Resources, A.H., M.N.A., E.F.A., A.H. and P.A.; Software, E.F.A., G.S. and M.B.S.; Supervision, S.A., M.R., M.B.S., M.A. (Muhammad Afzal) and A.H.; Validation, M.J.S. and A.H.; Writing—Original draft, N.N., A.H. and P.A.; Writing—Review and editing, S.A., M.A. (Muhammad Arslan), A.H., M.N.A. and P.A. All authors have read and agreed to the published version of the manuscript.

**Funding:** This research was supported by the Higher Education Commission, Pakistan (grant number 20-3854/R&D/HEC/14). The authors would like to extend their sincere appreciation to the Researchers Supporting Project Number (RSP-2019/116), King Saud University, Riyadh, Saudi Arabia.

**Acknowledgments:** The authors would like to appreciate the Government College University, Faisalabad, Pakistan for support. The authors would like to extend their sincere appreciation to the Researchers Supporting Project Number (RSP-2019/116), King Saud University, Riyadh, Saudi Arabia.

**Conflicts of Interest:** The authors declare that they have no conflict of interest.

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
