# Peer review of "Bacterial Augmented Floating Treatment Wetlands for Efficient Treatment of Synthetic Textile Dye Wastewater"

_sustainability, doi:10.3390/su12093731_

Round 1

Reviewer 1 Report

I have two small comments:

First, the authors didn't give some details that I ased for the wetlands (Is the flow subsurface or vertical? Which are the types of porous materials? Give values of porosity and conductivity).

Second, 92 references are too much, even in review papers we don't see this...

I believe htat this paper could be published, as the authors follow most of my comments/remarks, but if they want to increase the quality of their work, they could give the detalis of the CWs that I demand and remove about 30 references...

Author Response

Reviewer 1:

1- First, the authors didn't give some details that I asked for the wetlands (Is the flow subsurface or vertical? Which are the types of porous materials? Give values of porosity and conductivity).

Thank you very much. It is clear from the title of manuscript that we applied Floating Wetland in this research work. Page 2, Line 28-40 briefly describe the basic structure and function of floating wetlands. To make it more clear a new sentence (Line 38-40, page 2) has been added: “Contrast to conventional wetlands, floating wetlands can be installed on any aquatic pond without digging, earth moving and additional land acquisition”.

Further, Fig 1 at page 3, clearly describe the structure and functioning of the floating wetlands. Additionally, page 4, section 2.4, elaborates the fabrication process of floating wetlands and lines 17-20 in this section has been added about the porosity and conductivity of floating material: “Expanded Polystyrene (EPS) based sheets manufactured by Diamond® Foam Private Ltd. Pakistan. The features of EPS sheets are rigid, low thermal conductivity, moisture resistant and non-porous closed cell foam”.

2) Second, 92 references are too much, even in review papers we don't see this...I believe that this paper could be published, as the authors follow most of my comments/remarks, but if they want to increase the quality of their work, they could give the detalis of the CWs that I demand and remove about 30 references...

Detail of new information about CWs has been mentioned in comment 1. All the references have been readjusted and as per suggestion 30 references has been removed from this manuscript.

Reviewer 2 Report

You must indicate why you chose the indicated dyes: use, toxicity, etc.

Pages 6, 7 and 8. Figures footnotes 2, 3 and 4 must indicate to what dye are you refering to. Additionaly, can you explain the lettering? I understand that different letters means significant differences but what is the meaning of bars with several letters? The same applies for the Tables. 

Discussion, 2nd paragraph, 7th line. Please correct "wetland lands"

Many sentences mus be re-written. For instance, you did not measure organic acids so you cannot assure that pH was reduced by their release through the roots.

Color removal. You say that the FWT reduced dissolved oxygen. Did the water became anoxic? It is well known that many dyes are rapidly removed in anaerobic conditions. Can it be a mechanism in this case? Additionally, could pH be reduced by anoxic conditions instead of the release of organic acids?

pH reduction: you did not measure organic acids, how can you be so sure that the cause is that?

To show the persistance of inoculated bacteria the results from RFLP and electrophoresis patterns should be shown.

Author Response

Reviewer 2:

1) You must indicate why you chose the indicated dyes: use, toxicity, etc.

Thank you very much for your suggestion. Section 2.1, Page 3, Line 16-18 has been added to describe the reason of selection of these specific dyes: “These dyes were selected because of their common use in textile industry and high concentration of these toxic dyes and associated degraded products in textile effluent”.

2) Pages 6, 7 and 8. Figures footnotes 2, 3 and 4 must indicate to what dye you are referring to. Additionally, can you explain the lettering? I understand that different letters means significant differences but what is the meaning of bars with several letters? The same applies for the Tables.

In foot note of Fig. 2 (page 6), Fig 3 (page 7), Fig 4 (page 8) and table 1 (page 10), table 2 (page 11), table 3 (page 11), table 4 (page 12) the dye name has been indicated.

In this study a pairwise comparison was made between treatments x time. The treatments with several letters means these treatments are not significantly different from treatments sharing same letters. For example in table 1, D1 for Cu, T2 at 20 days is 65.9efg is not significantly difference from T2 at 10 days (58.5e) and T3 at 10 and 20 days (7.5efg and 75.0g), sharing same letter bar. As one treatment x time pair is not significantly different from more than two other pair, then it may have 3 or more alphabets on it.

3) Discussion, 2nd paragraph, 7th line. Please correct "wetland lands"

It has been corrected, Page 12, line 17.

4) Color removal. You say that the FWT reduced dissolved oxygen. Did the water became anoxic? It is well known that many dyes are rapidly removed in anaerobic conditions. Can it be a mechanism in this case? Additionally, could pH be reduced by anoxic conditions instead of the release of organic acids?

Thank you very much. The color removal process has been rewritten to end the confusion. Page 12, line 37-39 and page 13, line 1-7. Further line 19-21 page 12 has been removed to remove the confusion about FWT reduced dissolved oxygen.

5) pH reduction: you did not measure organic acids, how can you be so sure that the cause is that?

The page 12, line 10-11. The sentences has been corrected. You are absolutely right, we did not measure the organic acid. Here we described this pH reduction process in reference to previous studies who mentioned organic acids are responsible for pH reduction in floating wetlands.

6) To show the persistence of inoculated bacteria the results from RFLP and electrophoresis patterns should be shown.

Thank you for your suggestion. We are very sorry, we cannot do at this stage. However the data of microbial CFU has been presented in table 2 page 11.  

Round 2

Reviewer 2 Report

You provide no proof of the analyses of the inoculated bacteria. Thus, all the information regarding this point should be removed: en of Section 2.5, Section 3.3...

Author Response

Comments and Suggestions for Authors

You provide no proof of the analyses of the inoculated bacteria. Thus, all the information regarding this point should be removed: en of Section 2.5, Section 3.3...

Ans: Thank you very much for your suggestion. If we remove this information from these section it will badly damage the theme of the manuscript. As per your previous comment “To show the persistence of inoculated bacteria the results from RFLP and electrophoresis patterns should be shown”, we have added a picture at Page# 9 section 3.3 which represent the LB plates for CFU analysis electrophoresis pattern.

Here I would like to request you that, the most important thing related to bacterial strains is data about bacteria survival which is mentioned in table 2. Here, I would like to give you some examples from recently published research paper relevant to our study. In these research papers authors only mentioned data of bacterial survival. We humbly request you please withdraw this comment. We will be greatly thankful to for this kindness.

The research papers are following:

Tara, N., Iqbal, M., Mahmood Khan, Q., & Afzal, M. (2019). Bioaugmentation of floating treatment wetlands for the remediation of textile effluent. Water and environment journal33(1), 124 134. https://doi.org/10.1111/wej.12383

Tara, N., Arslan, M., Hussain, Z., Iqbal, M., Khan, Q. M., & Afzal, M. (2019). On-site performance of floating treatment wetland macrocosms augmented with dye-degrading bacteria for the remediation of textile industry wastewater. Journal of cleaner production217, 541-548. https://doi.org/10.1016/j.jclepro.2019.01.258

Round 3

Reviewer 2 Report

Dear author,

as you mention, the persistence of the inoculated bacteria is an important point of the research. This is why you should provide more detailed information on the identification of the inoculated bacteria. 

The better results of the inoculated reactors can be caused just by the higher concentration of bacteria regardless of the type of bacteria inoculated. You must show that the bacterial populations of the inoculated reactors are statistically different from those of non inoculated ones. Otherwise you cannot assure that the better results are provided by the new bacteria instead of the increased number of bacteria.

A figure showing microbial petri dishes and electrophoresis results do not demonstrate the persistence of the inoculated bacteria. You must provide details of the genomic analyses and statistics used. Additionally, the results obtained by other researchers do not demonstrate yours.

The other option is to provide different possible explanations on the better results of the inoculated reactors considering the possible persistence of the inoculated bacteria, the higher concentration of bacteria in the inoculated reactors, etc. But in my opinion, for the moment you have not shown the persistence of the inoculated bacteria.

Author Response

Reviewer 2

Comments and Suggestions for Authors

Dear author,

As you mention, the persistence of the inoculated bacteria is an important point of the research. This is why you should provide more detailed information on the identification of the inoculated bacteria. 

The better results of the inoculated reactors can be caused just by the higher concentration of bacteria regardless of the type of bacteria inoculated. You must show that the bacterial populations of the inoculated reactors are statistically different from those of non inoculated ones. Otherwise you cannot assure that the better results are provided by the new bacteria instead of the increased number of bacteria.

A figure showing microbial petri dishes and electrophoresis results do not demonstrate the persistence of the inoculated bacteria. You must provide details of the genomic analyses and statistics used. Additionally, the results obtained by other researchers do not demonstrate yours.

The other option is to provide different possible explanations on the better results of the inoculated reactors considering the possible persistence of the inoculated bacteria, the higher concentration of bacteria in the inoculated reactors, etc. But in my opinion, for the moment you have not shown the persistence of the inoculated bacteria.

Ans: Thank you very much for your suggestion. As per your suggestions, now we have added the data of bacterial population in water (Table 2), root and shoot (Table 3). The newly added data is highlighted blue. These tables show that population of bacteria in inoculated treatments is significantly higher than non-inoculated treatments. Further, we have rewritten the Section 2.5 (Page 5) and Section 3.3 (Page 9). We have completely removed the line about application of RFLP and Gel electrophoresis and Figure 6 which was about microbial petri dishes and electrophoresis resutls. As per your suggestion, we have modified the discussion and linked the better results of the inoculated reactors considering the possible persistence of the inoculated bacteria and higher population in inoculated reactors/treatments (Page 14, Line 40-46, 49-50) and Page 15 (Line 1-2).

Round 4

Reviewer 2 Report

I still consider that you have not proven that the innoculated bacteria persisted. The higher concentration of bacteria in the inoculated reactors does not mean that those colonies are the innoculated ones. The LB broth is a general one and does not help to identify the innoculated bacteria.

The paper has not been re-written according to my suggestions. 

In my opinion it should be rejected.

This manuscript is a resubmission of an earlier submission. The following is a list of the peer review reports and author responses from that submission.

Round 1

Reviewer 1 Report

  • The Abstract (and throughout the text) should provide % removals of the measured pollutants.
  • How was the synthetic textile wastewater prepared? Just tapwater with the dyes? With 50 ppm of a dye dissolved in tap water, you got a total suspended solid concentration of 92 ppm?
  • How do you explain EC reduction? Evapo-transpiration would increase rather than decrease EC.
  • Statistics. You say that you used Anova. Were the data homocedastic and normally distributed?
  • Please explain better the lettering employed to express statistical differences. I guess that different lettering on the bars means statistical differences but why do you use up to 4 letters, for example in Fig. 3A.
  • Section 3.2. You provide the final concentrations of heavy metals but not the removals. Please do it.
  • Section 3.3. How can you be so sure that the bacteria measured are the inoculated and not any other strain? You employed a generalistic incubation method. Didn't you?
  • Section 3.4. Please provide plant growth data for the experiments and percentages. That will make the paper easier to read.
  • Table 4. The data provided are concentration, removals?
  • Table 2. These data are supposed to be bacteria concentrations. Please provide units (CFU/100 mL?)
  • The Discussion section can be regarded as an Introduction. You provide many general comments but do not go deep into interpreting your results, contrasting with other studies, etc.

Reviewer 2 Report

BRIEF SUMMARY- BROAD COMMENTS

It is obvious that in this manuscript the authors present a big quantity of work. Despite this, it has many weaknesses that the authors should correct, in order this work to be considered ready for publication in “Sustainability” journal.

SPECIFIC COMMENTS

  1. Page 1, affiliations No. 5 and 7: Unite the words “Saudi” and “Arabia” and “2460,” and “Riyadh”, in order to be in the same lines.
  2. Page 1, abstract: Explain the exact meaning of each parameter in abstract, not in page 5. Thus, write “Electrical Conductivity (EC), Total Dissolved Solids (TDS)” exc..
  3. Page 2, Introduction: Give the original names of each parameter, i.e., “Copper (Cu)” exc..
  4. Page 3, Figure 1: This is a very bad-written part…Transfer the image of Figure 1 immediately before its explanation.
  5. Page 3, materials and methods: Give more details concerning the wetlands: Is the flow subsurface or vertical? Which are the types of porous materials? Give values of porosity and conductivity.
  6. Page 4, paragraph 2.3: Avoid words like “we” or “us” or “our”, as is sounds selfish. Delete “We applied” and write the first sentence like this: “A consortium of three bacterial strains was applied, namely….”.
  7. Page 5, Figure 2: This is a very bad-written part…Transfer the image of Figure 2 immediately before its explanation.
  8. Page 6, results: You must present the removal (in %) of each pollutants, after the operation of the wetlands.
  9. Pages 7-9, Figures 3-5: In the explanation of these figures, it is not necessary to repeat the meaning of each parameter. You could just write “(A) EC, (B) TDS, (C) TSS, (D) COD, (E) BOD”.
  10. Page 10: Did the wetlands operate in open-air facilities? If yes, which are the effects of weather conditions (rainfall, evapotranspiration exc.)?
  11. Page 14, conclusion: Replace “Our study” with “The present study”.
  12. Pages 14-15, conclusions: So far the authors didn’t persuade me about the importance of their work, as all their results could be found in many other studies…At least in the conclusions the authors should point the originality of their work and their contribution to science.
  13. REFERENCES: Check the style of the journals, from the directions, and make sure you present the references in the proper style. Check very carefully ALL the references if they are correct (names of authors and journals, titles exc.). For example, the best way is this of Ref. No. 10, where only the first word of the title begins with capital. Follow this for other studies, like No. 1, where you must write “Water pollution: Effects, Control and Management”. This is the proper way for journals. Repeat this for References No. 1, 2, 6, 9, 11, 21, 23. 32 and 58. Make sure that the whole reference is in the same page, and not splitted in two different pages, like No. 30 and 50. Finally, when there are conferences, you must give the whole name of the conferences (No. 46 and 59).

I believe that, only if the authors follow ALL my corrections, this

paper could suitable for Sustainability Journal.